# IMPORTANCE-BASED MULTIMODAL AUTOENCODER

## ABSTRACT

Integrating information from multiple modalities (e.g., verbal, acoustic and visual data) into meaningful representations has seen great progress in recent years. However, two challenges are not sufficiently addressed by current approaches: (1) computationally efficient training of multimodal autoencoder networks which are robust in the absence of modalities, and (2) unsupervised learning of *important* subspaces in each modality which are correlated with other modalities. In this paper we propose the IMA (Importance-based Multimodal Autoencoder) model, a scalable model that learns modality importances and robust multimodal representations through a novel cross-covariance based loss function. We conduct experiments on MNIST-TIDIGITS, a multimodal dataset of spoken and image digits, and on IEMOCAP, a multimodal emotion corpus. The IMA model is able to distinguish digits from uncorrelated noise, and word-level importances are learnt that correspond to the separation between function and emotional words. The multimodal representations learnt by IMA are also competitive with state-of-the-art baseline approaches on downstream tasks.

## 1 INTRODUCTION

With the ever-increasing amount of heterogeneous multimedia content present on the internet, machine learning approaches have been applied to automated perception problems such as object recognition (Krizhevsky et al., 2012), image captioning (Vinyals et al., 2015) and automatic language translation (Choi et al., 2018). An important research direction is the problem of learning representations from multiple modalities which *represent our primary channels of communication and sensation, such as vision or touch* (Baltrušaitis et al., 2018). With respect to this area of research, there are two major challenges in this research area which our paper addresses. The first is the design of encoder networks to enable learning and inference of multimodal representations in the absence of modalities. This is useful for scenarios such as sensor failure or imputation/bidirectional generation of missing modalities from any combination of the observed ones. The caveat is that to have this property, recent approaches such as the JMVAE-KL model (Suzuki et al., 2016) and MVAE (Wu & Goodman, 2018) have encoders with high complexity for a large number of modalities. When $M$ is the number of modalities, JMVAE-KL needs $2^M$ sub-networks for every combination of input modalities, while MVAE requires $M$ sub-networks but additional $O(2^M)$ subsampled loss terms to handle missing modalities.

The second challenge is that multimodal data, such as emotional spoken utterances or web images with captions are often generated not only by an underlying shared latent factor, but also by modality-specific private latent factors. For example in spoken utterances, the verbal modality (words) are generated not only due to emotion but also due to syntax and semantics. Function words such as *I* and *the* are mostly syntactic and do not relate to emotion, similarly not all recorded audio frames are indicative of emotion. The inference of how relevant a sample in each modality is to the shared latent factor (subsequently referred to as *importance*) is important for downstream tasks. For the remainder of this paper, non-relevant samples are refered to as *uncorrelated noise*.

In a supervised setting, the latent factors and modality importance weights can be learnt from task labels. When labels are absent in the unsupervised scenario, for the purpose of this paper we define the concept of *modality importances* based on correlations between the latent factor and each modality. In the important subspace of each modality the multimodal and unimodal representations both maximally correlate, indicating that samples in that modality subspace can be attributed to a

shared latent factor, and not an independent private one. In contrast, for unimportant samples in a modality, the correlation is minimal.

The main contributions of our proposed approach are two-fold. **The first** is a multimodal autoencoder framework where training requires additional loss terms which are $O(M)$, i.e. linear in the number of modalities, and thus only require $M$ per-modality encoders to handle missing modalities. Computationally, this is advantageous compared to JMVAE-KL and MVAE, which require exponential number of sub-networks and loss terms respectively. **Secondly**, we define the concept of importance in an unsupervised setting, and propose novel cross-correlation based loss terms to learn important regions in each modality's representation space. The importances are modeled by separate unimodal networks referred to as *importance networks*. Hyper-parameter $\rho_j$ for the $j$-th modality controls the integration of prior domain knowledge about the degree of importance in that modality. While not trained on any supervised labels, the learnt importances from IMA are analyzed quantitatively and found to correspond to the separation between digit vs. noise labels and emotion vs. neutral categories.

## 2 RELATED WORK

Following the great success of deep neural networks for representation learning, the research area of multimodal machine learning is gaining interest (Baltrušaitis et al., 2018). Our proposed IMA model is relevant to two main research areas in this domain, *Inter-modality Correlation Learning* and *Efficient Multimodal VAEs*. The idea of learning acoustic embeddings for words has also been explored in Wang et al. (2018) and Jung et al. (2019) however we attempt to map words to their affective rather than phonetic representations. In this section, we describe each area and conclude with the similarities and differences between the IMA model and prior approaches.

*Inter-modality Correlation Learning*: There have been several approaches which measure correlations between modalities/sources of data to understand how observed data in each modality can be explained by shared underlying concepts. The IBFA (Inter-Battery Factor Analysis) introduced by Tucker (1958) and its successor, the MBFA (Multi-battery factor Analysis) (Browne, 1980) are among the earliest proposed techniques to study shared factors between score sets from batteries of tests. DeepCCA (Deep Canonical Covariance Analysis) proposed by Benton et al. (2017) learns a deep projection of each modality in a bimodal understanding scenario so that the projections are maximally correlated, effectively extending the classical CCA technique (Knapp, 1978) to deep neural networks. Our proposed model extends these approaches to also detect important regions of each modality correlated with the shared latent factor.

*Efficient Multimodal VAEs*: VAEs (Variational Autoencoders) have been applied to multimodal data for applications such as inference and bidirectional generation of modalities. This poses a major challenge of constructing encoders to model the latent posterior which are efficient in training/inference under any combination of input modalities. Recent work addresses this by focusing on factorized models for efficient inference. Vedantam et al. (2017) employs a product-of-experts decomposition with modality specific inference networks to train image generation models. Wu & Goodman (2018) propose MVAE (Multimodal Variational Autoencoders), where the latent posterior is modeled with a parameter shared product of experts network. Shi et al. (2019) proposed a mixture-of-experts multimodal variational autoencoder (MMVAE) where the posterior is a mixture of experts instead. These approaches have been extended more recently, for example in multi-source neural variational inference (Kurle et al., 2019) where the multimodal posterior is constructed using learnt and integrated beliefs from multiple posteriors, each being informed by a different source. Sutter et al. (2020) introduce a novel Jensen-Shannon divergence based objective function which can be used to approximate both unimodal and joint multimodal posteriors.

While existing approaches attempt to efficiently learn multimodal representations through posterior modeling, our proposed IMA model aligns modalities during autoencoder training for projection to a common space which facilitates inference even in absence of modalities. Only $M$ encoders and $O(M)$ loss terms are required by the IMA model for inference with $M$ modalities. Prior work has also not focused sufficiently on unsupervised learning of modality importances (through detection of subspaces maximally correlated with shared latent factors) which we address in this paper.

## 3 MODEL DESCRIPTION

The IMA model consists of two main components : (1) the multimodal autoencoder and (2) the unimodal importance networks. In Figure 1 we have provided an overview diagram of the proposed model, including the loss functions utilized in training.

**Multimodal Autoencoder:** Assume that the input training examples consist of multimodal data, where each multimodal sample is denoted as $\mathbf{x} = \{\mathbf{x_1}, \mathbf{x_2}, ...\mathbf{x_M}\}$. There are $M$ modalities and the training set consists of $N$ multimodal samples. The input data $\mathbf{x_j}$ in the $j$-th modality is passed through an encoder for that modality and its output is denoted as $\mathbf{u_j}(\mathbf{x_j})$. The latent multimodal representation $\mathbf{z}$ could be modeled with different approaches for example, concatenated fusion with fully connected layers (Suzuki et al., 2016). We wish to model the latent multimodal representation $\mathbf{z}(\mathbf{x_1}, \mathbf{x_2}...\mathbf{x_M})$ with a pooling weighted by distinct modality importances $y_j(\mathbf{x_j})$ as given by:

$$\mathbf{z}(\mathbf{x_1}, \mathbf{x_2}, ...\mathbf{x_M}) = \sum_{j=1}^{M} y_j(\mathbf{x_j})\mathbf{u_j}(\mathbf{x_j}) \qquad \sum_{j} y_j(\mathbf{x_j}) = 1 \qquad (1)$$

The multimodal representation $\mathbf{z}$ is passed through the decoder networks to obtain the reconstruction in the $j$-th modality as $\hat{\mathbf{x}}_j$. $\mathcal{L}_{rec}^{(j)}$ is the reconstruction loss for the $j$-th modality. $\mathcal{L}_{glob}$ is a global regularization term which encourages $\mathbf{z}$ to be centered with zero mean. This term is for regularization of the multimodal representation $\mathbf{z}$ and expands to $\mathcal{L}_{glob}(\mathbf{z}) = \|\mathbf{z}(\mathbf{x_1}, \mathbf{x_2}, ...\mathbf{x_M})\|^2$.

**Multimodal alignment in IMA** : For our proposed model, we have $M$ sub-networks for each modality, where each $j$-th unimodal sub-network is trained using the autoencoder reconstruction losses and an additional alignment loss $\mathcal{L}_{align}^{(j)}$. By forcing the multimodal representation $\mathbf{z}$ and each modality's view $\mathbf{u_j}(\mathbf{x_j})$ to be similar, we also enforce each unimodal sub-network to learn its contribution to the latent factor $\mathbf{z}$ during autoencoder training. Per-sample, this requires only $M$ sub-networks and $M$ additional loss terms instead of a random subset of $2^M$ losses for sub-sampled training as in the MVAE model (Wu & Goodman, 2018) or $2^M$ subnetworks as in the JMVAE-KL model (Suzuki et al., 2016). Similar to MVAE, this model also can learn and infer in the absence of modalities. $\mathbf{z}$ can be expressed as a linear average of the unimodal representations present and incorporated into the overall loss. If the $j$-th modality is missing, we can set $y_j(\mathbf{x_j}) = 0$ in Equation 1. For the $j$-th modality, $\mathcal{L}_{align}^{(j)}$ is the SSE error between the multimodal representation $\mathbf{z}$ and the unimodal encoder outputs $\mathbf{u_j}(\mathbf{x_j})$ given by: $\mathcal{L}_{align}^{(j)}(\mathbf{z}, \mathbf{u_j}) = \|\mathbf{z}(\mathbf{x_1}, \mathbf{x_2}, ...\mathbf{x_M}) - \mathbf{u_j}(\mathbf{x_j})\|^2$ When there are $N$ training samples, during multimodal autoencoder learning the loss below is optimized:

$$\mathcal{L}_{auto} = \lambda_{glob}\sum_{i=1}^{N}\mathcal{L}_{glob}(\mathbf{z_{ij}}) + \left[\sum_{i=1}^{N}\sum_{j=1}^{M}\lambda_{rec}^{(j)}\mathcal{L}_{rec}^{(j)}(\mathbf{x_{ij}}, \hat{\mathbf{x}}_{\mathbf{ij}})\right] + \left[\sum_{i=1}^{N}\sum_{j=1}^{M}\lambda_{align}^{(j)}\mathcal{L}_{align}^{(j)}(\mathbf{z_i}, \mathbf{u_{ij}})\right] \qquad (2)$$

$\lambda_{align}, \lambda_{rec}$ and $\lambda_{glob}$ are the associated hyper-parameter weights for the alignment, reconstruction and global multimodal regularization terms which appear in Equation (2). Since it is not straightforward to tune such parameters for an unsupervised model, we start with equal weights for all losses and observe if that is sufficient to minimize all terms simultaneously during training. For the MNIST-TIDIGITS and IEMOCAP experiments, we have found that equally weighing them works, thus $\lambda_{align} = \lambda_{rec} = \lambda_{glob} = 1.0$.

**Importance Network Training:** Section 1 explains that there are samples in observed data for each modality which do not correlate with the multimodal representation $\mathbf{z}$. For the scope of this paper, we consider such samples as *uncorrelated noise* in the data. We have assumed the presence of a subspace $\mathcal{R}_j$ inside each $j$-th modality's representation space so that if $\mathbf{x_j} \in \mathcal{R}_j$, $\mathbf{u_j}$ is minimally correlated with $\mathbf{z}$. We seek to learn weights $y_{ij} \in [0, 1]$ for the $i$-th sample in the $j$−th modality, so that $y_{ij}$ denotes the importance of each sample $\mathbf{x_{ij}}$ (i.e. the degree to which $\mathbf{x_{ij}}$ does not belong to $\mathcal{R}_j$). A unimodal neural network is trained to map from $\mathbf{x_{ij}}$ to $y_{ij}$ for the $j$-th modality; this is called the *importance network*. The multimodal autoencoder solely being trained to reconstruct all modality inputs would not have learnt $y_{ij}$ corresponding to uncorrelated noise. This motivates the need for explicitly defining an importance-based loss.

**Cross-correlation Losses:** To train the importance networks, we have made use of a loss function explicitly capturing cross-correlation between $\mathbf{z}$ and $\mathbf{u_j}$, weighted by $y_j$. Minimizing this cost is equivalent to enforcing zero-correlation between $\mathbf{u_j}$ and $\mathbf{z}$ based on mini-batch statistics during training, where size of a mini-batch is $B$ samples. This cost can be derived from the definition of cross-correlation, where each $i$-th sample in the mini-batch is not weighted equally. Recent work

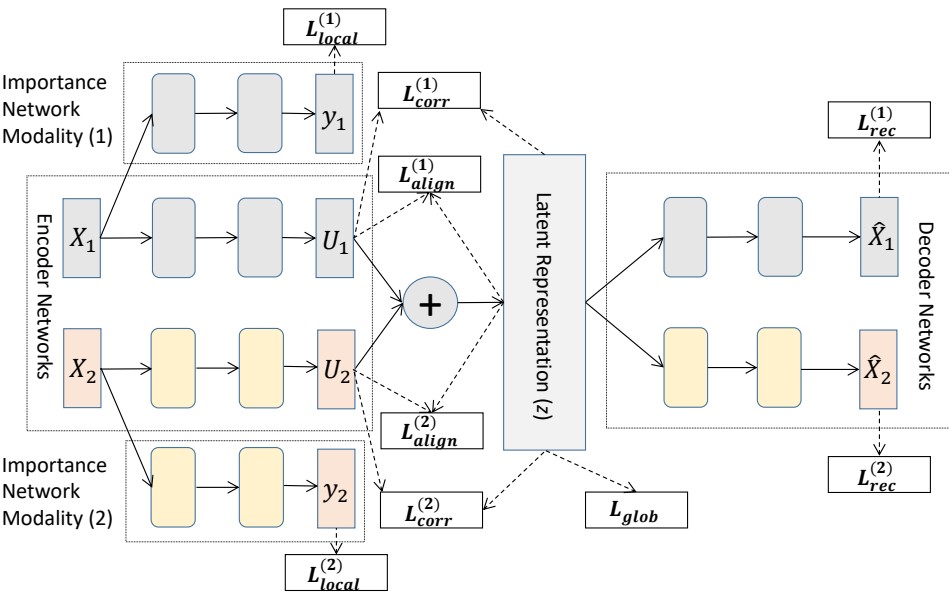

Figure 1: Overview of the proposed model IMA, including the multimodal autoencoder, the importance networks and the main loss functions to be optimized

has also applied other independence criterion, such as MMD (Maximum Mean Discrepancies) which act as auxiliary losses in a variational framework (Louizos et al., 2015). Assuming the latent variables are Gaussians, independence and uncorrelated properties are equivalent, and we define the following alternative loss term based on the Frobenius norm of the cross-covariance between $\mathbf{z}$ and $\mathbf{u_j}$, where $y'_{ij} = 1 - y_{ij}$ and $w'_{ij} = y'_{ij} \Big/ \sum_{i=1}^{B} y'_{ij}$

$$\mathcal{L}_{corr}^{(j)} = \left\| \sum_{i=1}^{B} w'_{ij} \left[ (\mathbf{z} - \mu_{\mathbf{z}})(\mathbf{u_j} - \mu_{\mathbf{u}}) \right]^T \right\|_F^2 \qquad \mu_{\mathbf{z}} = \sum_{i=1}^{B} w'_{ij} \mathbf{z_i} \qquad \mu_{\mathbf{u}} = \sum_{i=1}^{B} w'_{ij} \mathbf{u_{ij}} \qquad (3)$$

**Importance Priors:** Learning to predict $y_{ij} \approx 1.0 \ \forall i, j$ trivially decreases $\mathcal{L}_{corr}$ down to zero, and thus we need to regularize the importance network training, through an additional loss function which we refer to as $\mathcal{L}_{local}^{(j)}$. This loss utilizes the hyper-parameter $\rho_j \in [0, 1]$ which serves as prior about how much of the $j$-th modality is corrupted by uncorrelated noise, and is defined as the KL-divergence $D_{KL}(\text{Bernoulli}(\bar{y}_j) \| \text{Bernoulli}(\rho_j))$ where $\bar{y}_j = \sum_i y_{ij}/B$ is the average value of $y_{ij}$ as computed over a mini-batch of size $B$. Enforcing the loss $\mathcal{L}_{local}$ at a sample level would have forced $y_{ij} = \rho_j$ for all samples, which is addressed by defining the loss on a mini-batch instead. The importance network for each modality minimizes $\mathcal{L}_{imp}$, which is the weighted sum of the cross-covariance based loss and the regularization term, as defined below. $\lambda_{local}^{(j)}$ and $\lambda_{corr}^{(j)}$ are hyper-parameter weights for each loss term.

$$\mathcal{L}_{imp} = \sum_{i=1}^{N} \sum_{j=1}^{M} \lambda_{local}^{(j)} \mathcal{L}_{local}^{(j)} + \sum_{i=1}^{N} \sum_{j=1}^{M} \lambda_{corr}^{(j)} \mathcal{L}_{corr}^{(j)} \qquad (4)$$

It is important to note the difference between $y_{ij}$ and $y'_{ij} = 1 - y_{ij}$ in terms of notation. $y_{ij}$ is the importance of the $i-$th sample in the $j-$th modality and is required in the joint representation, whereas its complement $y'_{ij}$ is utilized in importance network training. The intuition is that a model minimizing Equation 3 would tend to learn high values of $y'_{ij}$ for maximally uncorrelated samples, which translates to low importance ($y_{ij}$) for these same samples.

## 4 DATASETS

We perform experiments with the proposed IMA model on (1) MNIST-TIDIGITS which is a multimodal digit dataset created by pairing image and spoken samples from common image and speech digit datasets and (2) the IEMOCAP corpus (Busso et al., 2008). MNIST-TIDIGITS is synthesized

from three datasets for digit recognition from images and speech - MNIST dataset (Salakhutdinov & Hinton, 2007); TIDIGITS connected spoken digit sequence corpus (Leonard, 1984) and TI46 digits (Liberman, 1993). We have merged these two spoken digit datasets to create MNIST-TIDIGITS, a dataset as large as the number of samples in MNIST. The motivation underlying this is to pair image and spoken digits to create a multimodal dataset. USC-IEMOCAP is a well-known dataset for speech emotion recognition comprising of acted and spontaneous multimodal interactions of dyadic sessions between human actors. The dataset consists of around 12 hours of speech from 10 human subjects, and is labeled by three annotators for emotions such as *Happy*, *Sad*, *Angry*, *Excitement*, *Neutral* and *Surprise*. We train IMA model on all utterances, but perform classification/retrieval experiments only for the four standard primary emotion categories - *Neutral*, *Angry*, *Sad* and *Happy*, with a total of 5531 utterances belonging to these categories (Kim et al., 2013; Jin et al., 2015).

**MNIST-TIDIGITS:** We construct a multimodal dataset by pairing the MNIST digit images with the TIDIGIT spoken digits based on the common digit label. We define a parameter $R$ called the *noise factor* to control the amount by which each digit image or spoken digit co-occurs with synthesized noise samples. For each label category $c$ with $N$ image/spoken digits, we create three sets of multimodal samples: $N * R$ samples consisting of images of digit $c$ paired with synthetic spoken noise; $N * R$ samples consisting of synthetic image noise paired with samples of spoken digit $c$ and $N * (1 - R)$ samples consisting of images of digit $c$ paired with spoken digit $c$. For all multimodal samples, the ground truth labels are set to the actual digit labels, disregarding the synthetic noise. For example, for a multimodal sample with (image '7', spoken noise), the sample label is set to 7. Note that the sample label is not known at model training time.

**IEMOCAP:** Consider each IEMOCAP spoken utterance to have $N$ word tokens and word-level acoustic feature vectors (for example, descriptors such as pitch). We pair each token with all $N$ acoustic vectors in the same utterance, creating $N$ paired multimodal samples. Thus if an IEMOCAP utterance initially has $N$ word-acoustic pairs, it is expanded to $N^2$ pairs for model training. We also remove the MFCC features, and thus most of the local phonetic variability within the utterance is removed resulting in global factors of variation, such as emotion. The scope of the non-verbal acoustic features co-occurring with a word is expanded to the entire utterance. This is analogous to techniques such as skip-gram word embeddings (Mikolov et al., 2013) where a word context spans beyond adjacent words, resulting in an expansion of word pairs to train on.

## 5 EXPERIMENTAL SETUP

In this section, we describe the experimental setup including downstream experiments to evaluate the quality of the multimodal representations learnt by the proposed IMA model. The model training is performed in two stages. In the first stage the importance network weights are kept constant, and the multimodal autoencoder is trained. In the second stage the modality-specific importance networks are trained; with multimodal autoencoder weights kept constant. This sequence of alternating optimization could be performed for a finite number of iterations, but in practice we have found one stage to be sufficient. The image and speech modality inputs for the MNIST-TIDIGITS datasets have 784 and 696 dimensions respectively. The two-layer encoder and decoder networks for the multimodal autoencoder have 500 neurons/layer for image and 100 neurons/layer for the speech modality. A binary cross-entropy loss is used for autoencoder training. For IEMOCAP, the verbal embeddings (GLOVE) and acoustic features have 200 and 55 dimensions respectively. The two-layer autoencoder has 100 neurons/layer in both modalities, and the SSE (Sum of Squared Error) objective is used for training.

### 5.1 IMPORTANCE PRIOR

We refer to the parameter $\rho_j \in [0, 1]$ as the *importance prior* for the $j$-th modality. $\rho_j$ can be adjusted to control the proportion of samples detected as uncorrelated noise and is related to domain knowledge. For example in IEMOCAP, $\rho$ for text can be set to a very low value, since we hypothesize that a vast majority of vocabulary words (e.g. function words) do not correlate with the latent factor (such as emotion). Similarly for the MNIST-TIDIGITS dataset, the amount of uncorrelated noise (the aforementioned *noise factor* in Section 4) in the paired multimodal dataset is 10%, requiring a high value of $\rho_j$ since $\sim 90\%$ of samples in each modality correspond to valid digits.

## 5.2 EVALUATION OF IMPORTANCE NETWORK PERFORMANCE

Since the model is unsupervised where the actual importances are not available for training, there is no quantitative way to determine the best value of the hyper-parameter $\rho_j$ for each problem. However we have utilized the ground truth labels from the MNIST-TIDIGITS and IEMOCAP datasets to evaluating performance of the modality-specific importance networks. For the $i$-th sample in the $j$-th modality, the importance network maps input $\mathbf{x_{ij}}$ to an importance score $y_{ij} \in [0, 1]$. Given the ground truth assignment of $x_{ij}$ to a positive or negative class depending on whether it is uncorrelated noise, we can compute F1 scores to determine the optimal importance prior parameter, as well as quantify importance network performance with varying values of $\rho_j$.

For the IEMOCAP dataset, the important verbal category is defined to be the class of all vocabulary words belonging to the LIWC emotional categories (Pennebaker et al., 2001), namely - *sad*, *angry*, *anxiety*, *negative emotion* and *positive emotion*. In the acoustic modality, the important category is defined to be the class of non-verbal acoustic samples belonging to the *happy*, *angry* and *sad* emotions as annotated in the IEMOCAP dataset. For MNIST-TIDIGITS, importance is indicated by the presence of actual images/spoken digits in any of the modalities. Uncorrelated synthetic noise is not important, and is expected to be detected by the importance networks.

## 5.3 MULTIMODAL REPRESENTATIONS

**MNIST-TIDIGITS:** IMA learns importances $y_{ij}$ for the $j$-th modality (images:$j = 0$/speech:$j = 1$) and for the $i-$th multimodal sample, if $\mathbf{u_{ij}}$ are outputs of unimodal encoders, the multimodal representation is obtained by: $\mathbf{z_i} = (y_{i0}\mathbf{u_{i0}} + y_{i1}\mathbf{u_{i1}})/(y_{i0} + y_{i1})$

**IEMOCAP:** IMA learns token-level importances from the co-occurrences of words (modality 0) and their paired acoustic feature vectors (modality 1), which we combine to produce utterance representations. Each $i$-th utterance of length $L$ comprises of a bag of words $\{w_{i1}, w_{i2}, \cdots w_{iL}\}$ and a bag of acoustic vectors $\{\mathbf{v_{i1}}, \mathbf{v_{i2}}, \cdots \mathbf{v_{iL}}\}$. Their respective unimodal representations are $\{\mathbf{u_0}(w_{i1}), \cdots \mathbf{u_0}(w_{iL})\}$ and $\{\mathbf{u_1}(\mathbf{v_{i1}}), \cdots \mathbf{u_1}(\mathbf{v_{iL}})\}$. Corresponding importances are $\{y_0(w_{i1}), \cdots y_0(w_{iL})\}$ and $\{y_1(\mathbf{v_{i1}}), \cdots y_1(\mathbf{v_{iL}})\}$. The multimodal representation of the whole utterance $\mathbf{s_i}$ is given by:

$$\mathbf{s_i} = \sum_{l=1}^{l=L} [y_0(w_{il})\mathbf{u_0}(w_{il}) + y_1(\mathbf{v_{il}})\mathbf{u_1}(\mathbf{v_{il}})] \Big/ \sum_{l=1}^{l=L} [y_0(w_{il}) + y_1(\mathbf{v_{il}})] \tag{5}$$

We also compare multimodal performance with representations obtained only from the unimodal encoders. In all cases, we also perform an ablation experiment which does not utilize the learnt importances $y_0$ or $y_1$ and computes an unweighted average of the unimodal representations.

## 5.4 RETRIEVAL EXPERIMENTS

We conduct an intrinsic evaluation of the multimodal representation quality through retrieval experiments. Even if a modality is corrupted through uncorrelated noise, the multimodal representation should filter it out by weighing it less. Ideally, two samples with the same label should have similar representations. We define the concept of relevance as: *Two multimodal samples are relevant if they correspond to the same ground truth label, even with noise in any of the modalities.* For each of $i \in \{1, 2, 3...N\}$, we obtain the $K$ nearest neighbors of the $i$-th sample in terms of the Euclidean distance and count the number of relevant neighbors as $C_i$. The Precision@K score is computed as $\text{Pr@K} = \sum_i C_i/NK$.

**MNIST-TIDIGITS:** Results are reported both for two and 50-dimensional representations at $K \in \{10, 50\}$. The ground truth label is the digit label. The representation of a (MNIST image digit, speech noise) pair should be similar to the representation of (image noise, spoken TIDIGITS digit) if the underlying digit labels are the same. **IEMOCAP:** The retrieval experiments on the IEMOCAP dataset are at utterance level, and consider the primary emotion categories as ground truth labels. For the retrieval experiments, we define two utterances to be relevant if *they occur in the same IEMOCAP primary emotion category, one out of: happy, angry, sad and neutral.* The Precision@K scores for $K = 10$ are reported for 100-dimensional representations.

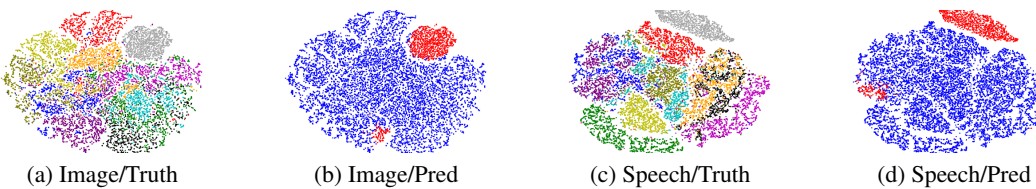

|                  |                 |                   |                  |
| :--------------: | :-------------: | :---------------: | :--------------: |
| (a) Image/Truth  | (b) Image/Pred  | (c) Speech/Truth  | (d) Speech/Pred  |

Figure 2: t-SNE visualizations of importance network representations learnt by the model on multimodal MNIST-TIDIGITS. Sub-figures (a) and (c) shows the digit clusters in the representations. The grey cluster corresponds to uncorrelated noise. In sub-figures (b) and (d) uncorrelated noise is in red; otherwise in blue.

```
you  are  far  too  temperamental  try  <sil>  to  control  yourself

<sil>  i  don't  really  have  any  imagination  <sil>  that's  all  i  know  how  to  <sil>  say  <sil>

it  certainly  is  not  <sil>  it's  <sil>  slightly  exaggerated  scientific  fact

very  well  <sil>  if  you  insist  on  being  <sil>  boorish  and  idiotic

they  didn't  die  they  <sil>  killed  themselves  for  each  other  <sil>  i  mean  that  exactly  <sil>  just
more  selfish  and  they  would  all  be  here  today  <sil>

i  don't  know  <sil>  when  it  cracked  he  ran  into  the  house  and  cried  in  the  kitchen
```

Figure 3: Word level importances learnt by the IMA model as observed on five example IEMOCAP utterances.

## 6 EXPERIMENTAL RESULTS

We have compared performances of the following models in our experiments: (1) Joint multimodal representations (unweighted and importance-weighted) from proposed IMA model as described in Section 5.3. (2) Unimodal representations from IMA model which just considers the output $\mathbf{u_j}$ of each of the $j$-th modality-specific encoders as the final representation. (3) MVAE model proposed in Wu & Goodman (2018) (4) JMVAE-KL model proposed in Suzuki et al. (2016).

### 6.1 IMPORTANCE NETWORKS

We evaluate the performance of the importance networks by evaluating the learned importances in each modality against domain knowledge based labels for uncorrelated noise presence.

**MNIST-TIDIGITS:** On MNIST-TIDIGITS we record precision, recall and F1 scores for values of $\rho \in [0.05 - 0.95]$. The best F1 score occurs at $\rho = 0.8$ and $0.3$, which are empirically the suitable priors for the importance network predictions to agree with external labels in the image and speech modalities respectively. For these optimal values of $\rho$ in each modality, we visualize the top layer of each importance network which implements $y_{ij}$ for the $j$-th modality given $\mathbf{x}_{ij}$. Figure 2(a-d) shows the t-SNE representations for MNIST-TIDIGITS, with both the ground truth label clusters and the importance network outputs. We observe that the network correctly learns the regions of the representation which correspond to uncorrelated noise. Around 2% of the data is misclassified by the importance network, which manifests in the small red region in Figures 2(b, d). The t-SNE visualizations for all the representations in the paper are obtained using a *perplexity* of 10.0 and a *delta* of 0.5.

**IEMOCAP:** We recorded F1 scores for the importance network with $\rho$ varying in $[0.0, 1.0]$ for two modalities: (a) vocabulary words and (b) acoustic non-verbal features. The evaluation is performed on the validation set. The F1 scores are computed for both the positive and negative classes in each modality, along with the average scores. The average F1 score for words peaks at $\rho = 0.03$ (value of 0.595), and for the acoustic modality at $\rho = 0.7$ (value of 0.508). Since a lot of vocabulary words do not have emotional information, $\rho$ at which the average verbal F1 score reaches maximum is a small value. We compare with a random classifier parameterized on $r$ which labels each sample as True with probability $r$ and False with probability $1 - r$, where $r \in [0, 1]$. For vocabulary words, the best average F1 score obtained was 0.552 at $r = 0.2$; and in the acoustic modality the best average F1 score obtained was 0.500 at $r = 0.1$. In both cases, the best F1 score obtained from the importance network outperforms that of the random baseline. We also qualitatively examine the word-level importances learnt by the IMA model for IEMOCAP. For the best performing value of $\rho$, Figure 3 shows five example utterances and for each utterance, the words are shaded from light to dark in order of increasing importance. Emotional words such as *temperamental*, *boorish* and *exaggerated* are detected as important by the model while function words (*to*, *the*, pause *'sil'* are ignored.

Table 1: Precision@K scores (P@K) corresponding to K=10 and 50 for the IMA model and baselines for the task of retrieval of similar multimodal MNIST-TIDIGITS paired data.

| Dim | Precision | Model | Multimodal | | Image (MNIST) | Speech (TIDIGITS) |
|---|---|---|---|---|---|---|
| | | | Weighted | Unweighted | | |
| 2D | Pr@10 | IMA | **0.8042** | 0.7649 | **0.7448** | 0.6180 |
| | | JMVAE-KL | 0.7392 | | 0.6080 | **0.6676** |
| | | MVAE | 0.6116 | | 0.4638 | 0.5285 |
| 2D | Pr@50 | IMA | **0.7812** | 0.7354 | **0.7180** | 0.5750 |
| | | JMVAE-KL | 0.7013 | | 0.5655 | **0.6298** |
| | | MVAE | 0.5655 | | 0.4090 | 0.4656 |
| 50D | Pr@10 | IMA | **0.9489** | 0.9414 | 0.8500 | 0.7788 |
| | | JMVAE-KL | 0.9013 | | **0.8618** | **0.8436** |
| | | MVAE | 0.9015 | | 0.8260 | 0.8071 |
| 50D | Pr@50 | IMA | **0.9326** | 0.8988 | **0.8240** | 0.7248 |
| | | JMVAE-KL | 0.7645 | | 0.7923 | **0.7309** |
| | | MVAE | 0.7666 | | 0.7139 | 0.6573 |

Table 2: Precision@K scores at K=10 for the proposed IMA model and other baselines for the task of same-emotion utterance retrieval from IEMOCAP (for *happy, angry, sad, neutral* emotions).

| Model | Verbal | | Acoustic | | Multimodal | |
|---|---|---|---|---|---|---|
| | Unwt | Wt | Unwt | Wt | Unwt | Wt |
| IMA | 0.3735 | **0.402** | 0.4634 | 0.4501 | 0.4781 | 0.4585 |
| JMVAE-KL | 0.3821 | | **0.4831** | | 0.5003 | |
| MVAE | 0.3874 | | 0.4759 | | **0.5045** | |

## 6.2 RETRIEVAL EXPERIMENTS

**MNIST-TIDIGITS:** Multimodal representations are computed by weighing unimodal image/speech digit representations with learned importances as described in Section 5.3. The IMA model would be expected to filter uncorrelated noise and map a paired combination of a (noise sample in one modality + a true digit in the other modality) to a cluster corresponding to the true digit. This property may not be observed in other approaches to multimodal representations including baselines (Suzuki et al., 2016; Wu & Goodman, 2018). Representation quality can be evaluated through retrieval experiments as explained in Section 5.4. Table 1 presents Precision@K scores (which are in the range [0,1]) at $K$=10 and 50 for the test set of MNIST-TIDIGITS. We compute precision results for the IMA model as well as the competing baselines, including a multimodal denoising autoencoder (with denoising rate=0.3). The importance-weighted multimodal representations outperform the JMVAE-KL and MVAE models (and also the unweighted multimodal representation from IMA) due to the model's ability to filter uncorrelated noise in each modality. The JMVAE-KL model does not weigh modalities, and while the IMA model does this when computing the product of experts, those weights do not take into account regions of each modality having uncorrelated noise. The denoising autoencoder obtains multimodal precisions (2D Pr@10: 0.5975 ; Pr@50: 0.5488 ; 50D Pr@10: 0.8968 ; Pr@50: 0.7818) which are worse than the IMA model. We also report performance when only one modality (either images or speech) is present at inference time. For the image modality the IMA model performs the best, while for speech JMVAE-KL provides the highest precision.

**IEMOCAP:** Section 5.3 describes how the importance-weighted unimodal and multimodal representations are obtained for each utterance, along with representations which do not consider learned importances. Table 2 shows the Precision@K scores on the validation set for the multimodal representation **z** obtained from the IMA model and the comparison baselines. For all approaches, the verbal representation performance is lower than acoustic non-verbal modality which is due to the GLOVE representations capturing semantic (and not necessarily emotional) information from words. The multimodal representations also have superior Precision@K scores compared to the unimodal performances. The improvement from learning importances is observed for the verbal modality, where it leads to a 2.67% improvement in the Precision@10 metric. This improvement occurs as the weighted utterance representations are obtained by filtering low importance (ex. func-

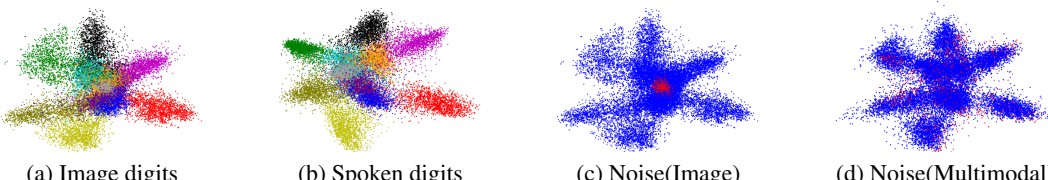

|  |  |  |  |
|---|---|---|---|
| (a) Image digits | (b) Spoken digits | (c) Noise(Image) | (d) Noise(Multimodal) |

Figure 4: t-SNE visualizations of unimodal representations learnt by the IMA autoencoder on MNIST-TIDIGITS. Colors denote digit labels (0:*Red*; 1:*Green*; 2:*Blue*; 3:*Purple*; 4:*Orange*; 5:*Cyan*; 6:*Yellow*; 7:*Magenta*; 8:*Olive*; 9:*Black*; Gray:*Uncorrelated Noise*). Sub-figures (c) and (d) show the region of image noise in red, superimposed on the image and multimodal representation spaces respectively.

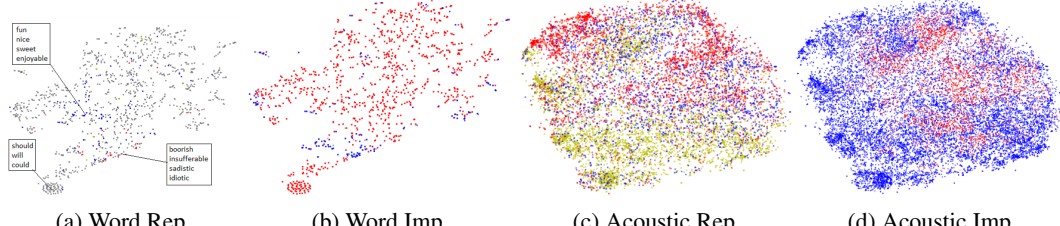

|  |  |  |  |
|---|---|---|---|
| (a) Word Rep. | (b) Word Imp. | (c) Acoustic Rep. | (d) Acoustic Imp. |

Figure 5: Visualization of different unimodal representations learnt by the IMA model on IEMOCAP. Figure 5(a) shows the unimodal word representations. Figure 5(b) shows the same representations colored with respective word importances. Figure 5(c) and (d) show the acoustic representations colored by emotion and importances respectively. Note that the colors blue, red and yellow respectively denote the *happy*, *angry* and *sad* emotions. The regions learned by IMA as *important* are in *blue*; *non-important* are in *red*.

tion words) out from the utterance. The weighted verbal representations from the IMA model outperform the JMVAE-KL and MVAE baselines on the Precision@10 metric, however corresponding scores are less on the acoustic and multimodal representations.

### 6.3 VISUALIZATION OF LEARNT REPRESENTATIONS

**MNIST-TIDIGITS :** In Figure 4(a, b), the unimodal representations $\mathbf{u_j}$ for $j \in \{1, 2\}$ are presented, along with the joint multimodal representation of the paired samples. For MNIST-TIDIGITS, $j = 1$ and $j = 2$ correspond to image and speech respectively. We note that the loss term $\mathcal{L}_{align}$ makes the unimodal representations super-imposable. Figure 4(c, d) shows the uncorrelated noise region in the image embedding space. They get mapped to their relevant digit locations in the multimodal space due to importance weighting (in this case $w_{speech} >> w_{image}$). **IEMOCAP:** Figure 5 shows t-SNE visualizations of representations learnt by the IMA model on IEMOCAP. Figures 5(a, b) show word representations, and we observe that the emotionally colored words are at the periphery and distinct from neutral words. The IMA model also detects most words as unimportant at the optimal value of $\rho$. These word representations are learnt from GLOVE embedding inputs to the model. The acoustic representations are in Figures 5(c, d). The regions which correspond to distinct emotion clusters such as *happy*, *angry* and *sad* are detected as important by the IMA model, and regions with neutral emotion and/or high confusion between emotions are assigned low importances.

## 7 CONCLUSIONS

We have proposed the IMA model for robust learning of multimodal representations. The proposed model performs unimodal inference in absence of modalities and also addresses the problem of detecting important subspaces in each modality through weighted cross-covariance loss terms, which are minimized by unimodal importance networks. The IMA model is trained on standard datasets such as MNIST-TIDIGITS as well as conversational datasets such as IEMOCAP. Experiments are performed to evaluate quality of the learned representations for downstream multimodal tasks such as digit recognition and emotion understanding. The IMA model achieves performance competitive with baseline approaches. Accuracy on these tasks is also improved by incorporating learnt unimodal importances into the multimodal representations. Future work could focus on extension to more than two modalities (such as visual), and sequence models such as RNNs (Recurrent Neural Networks) in encoder-decoder networks for utterance level analysis.

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

## A  APPENDIX

### A.1  MNIST-TIDIGITS Dataset Processing

The datasets which we have considered for our experiments are further pre-processed to create feature sets amenable to model training, these steps mainly involve:

1. Feature extraction from the MNIST images and the TIDIGITS/TI46 speech corpora.

2. Multimodal pairing for model to learn co-occurrences ie. images of spoken digits are paired with their respective spoken utterances to create multimodal paired samples.

3. Addition of samples with uncorrelated noise in each modality which co-occur with all image/speech digits in the other modality. The model is expected to learn that the noise samples in each modality are not important.

The composition of each noise image or speech sample is not important here; rather it is their co-occurrence pattern with the rest of the modalities which the model learns from during training. The only constraint we impose on the noise samples is the locality of their representations in the modality space to enable a learnt hyperplane to separate it out from the rest of the samples. We pre-process each image digit from MNIST and each spoken digit from TIDIGITS/TI46 to create 784 and 696-dimensional binary feature vectors respectively. We also construct synthetic image and spoken noise samples for each modality which serves as uncorrelated noise.

### A.1.1 PROCESSING OF MNIST AND NOISE DIGITS

Each MNIST digit is of 28x28 dimensionality, and the pixel intensities are in the range [0,255]. We divide each intensity by 255 to normalize the pixel values to the range [0,1]. We also construct images which are the same dimensionality (28*28) as the MNIST digits, but consist of white Gaussian noise at a predefined level of sparsity $K$. In our experiments, we have selected $K = 0.8$. The sparsity is introduced to make the synthetic noise compatible with the cross-entropy loss function. Synthesis is performed by creating a 28*28 grayscale image of white Gaussian noise at each pixel with an intensity mean of 0.5 and a standard deviation of 0.12, which ensures that almost all the image pixels are in the range [0,1], and then setting $K$-fraction of those pixels to zero. Negative intensity pixels are also set to zero. All images (MNIST and noise) are subsequently flattened, resulting in 784-dimensional feature vectors.

### A.1.2 PROCESSING OF TIDIGITS/TI46 SPOKEN DIGITS

The speech waveforms from the TIDIGITS and the TI46 datasets are first resampled to 12.5 KHz, and then the TIDIGITS dataset is further segmented by speaker and digit to generate audio files, each corresponding to a single speaker uttering one digit. Noise speech samples are generated by sampling white Gaussian noise for each time step with a mean of 0 and a standard deviation of 1; which corresponds to the same numeric amplitude ranges as the digit audio files. Each synthesized noise waveform is of one second duration and has a sampling rate of 12.5 KHz. Subsequently, MFCC (Mel Frequency Cepstral Coefficient) features are extracted from each audio file (both the synthesized noise and the spoken digits), with 20 ms windows; 10 ms shift and 12 cepstral coefficients not including the energy term. Each digit utterance also consists of recording pauses at the start and end time samples. To filter these pauses, 29 frames from either side (left and right) of the utterance midpoint are selected and concatenated to form a 58*12=696 dimensional feature vector; the remaining frames are discarded.

### A.2 IEMOCAP DATASET PROCESSING

We represent vocabulary words in the IEMOCAP dataset with their pre-trained embeddings using GLOVE (Global Vectors for Word Representation) (Pennington et al., 2014), a popular technique for word embeddings. For model training we have only considered vocabulary words which appear at least 5 times in the corpus, resulting in a vocabulary size of 1215 words. We use the COVAREP toolbox to extract features from speech waveforms in the IEMOCAP dataset. COVAREP is an open-source toolbox (Degottex et al., 2014) commonly used in voice-based analysis applications. Acoustic features are extracted at speech frame level to form a 55-dimensional feature vector, and a spoken word is represented by the mean of co-occurring frame vectors.

Acoustic signals incorporate multiple factors of variation such as phonetics (acoustic characteristics pertaining to what is said) as well as extra-linguistic factors (acoustic characteristics pertaining to how or by whom it is said) such as emotions (happy vs. angry voice, etc.), loudness, tone of voice, and gender. While all of these latent factors influence the observed acoustic features, they are both local as well as global in nature. Local factors of variation are transient, for example, phonetics (fricative, vowel sounds, etc.) and pause fillers. Global factors of variation influence the whole utterance, for example higher or lower average pitch (which can vary with gender, i.e., a global attribute).

The MFCCs mostly capture phonetics, and we have removed them as we do not wish to learn how words correlate with their phonetics, but rather to their emotional/extra-linguistic attributes. The phonetic latent factors would tend to dominate shared latent representation if MFCCs are included during training. We are seeking to learn latent extra-linguistic speech factors instead of phonetic components. The dataset also has transcriptions with token-level timestamps. For each spoken token, we average the COVAREP features extracted from the waveform co-occurring with it to create a single 55-dimensional acoustic feature vector. Each acoustic vector consists of features useful in voice analysis such as NAQ (Normalized Amplitude Quotient), QOQ (Quasi-Open Quotient) and F0 (Fundamental Frequency) with a sliding window of 20 ms. duration and 10 ms. shift.

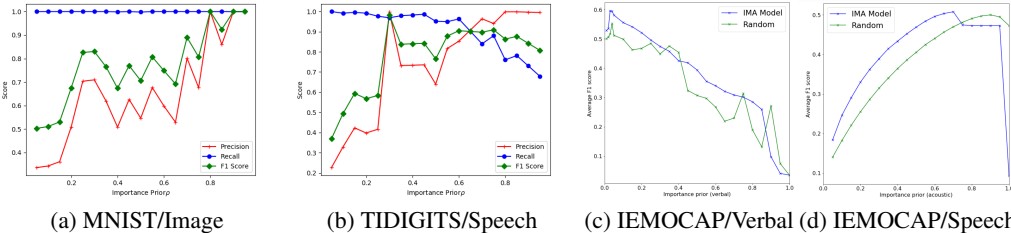

(a) MNIST/Image  (b) TIDIGITS/Speech  (c) IEMOCAP/Verbal (d) IEMOCAP/Speech

Figure 6: Precision, Recall and F1 score curves for the importance-based autoencoder trained on the MNIST-TIDIGITS dataset. We have selected only the Positive category (positive indicates presence of noise) for reporting metrics.

## A.3 IMPORTANCE NETWORK PERFORMANCE

In Figure 6, we present the variation of the precision, recall and F1 scores for the importance network performance on the MNIST-TIDIGIT and IEMOCAP datasets with varying values of the importance prior $\rho$ in the range $[0, 1]$. The scores explained in Section 6.1 correspond to these figures, and also include the comparison with a random baseline importance network for the IEMOCAP experiments.

Table 3: Experimental setup for digits (MNIST-TIDIGITS) and emotion classification (IEMOCAP)

| Parameters Dataset | Digits (MNIST-TIDIGITS) | Emotion (IEMOCAP) |
|---|---|---|
| Learning Rate ($eta$) range | {0.0625,0.125,0.25, 0.5,1.0,2.0,4.0,8.0} | {0.0625,0.125,0.25, 0.5,1.0,2.0,4.0,8.0} |
| Train set size | 50000 | 1115 |
| Validation set size | 15000 | 526 |
| Test set size | 10000 | - |
| Neurons/layer | 50 | 50 |

## A.4 MODEL SIZES OF IMA AND BASELINES

The IMA model has the advantage of having only $M$ encoders and $O(M)$ loss terms to optimize compared to the JMVAE-KL and MVAE models. That in conjunction with no stochasticity assumption (present in VAEs) results in fewer autoencoder parameters to train compared to JMVAE-KL and MVAE, even when the encoder and decoder sizes are the same. Table 4 lists the model sizes for the IMA model and the baseline models for the experimental settings described in Section 5. The IMA model has slightly higher parameters compared to MVAE, however that increase comes with the ability of the importance networks to learn uncorrelated noise across modalities.

Table 4: Number of parameters of IMA model compared with the JMVAE-KL and MVAE baselines

| Model Component | IMA | JMVAE-KL | MVAE |
|---|---|---|---|
| Unimodal Encoders | 753000 | 783000 | 783000 |
| Multimodal Encoders | 0 | 1309300 | 0 |
| Decoders | 754280 | 1217360 | 1217360 |
| Importance Network | 723402 | 0 | 0 |
| Total Size | 2230682 | 3309660 | 2000360 |

Table 5: Test set accuracies obtained by the multimodal embeddings from the proposed IMA model and other baselines on MNIST-TIDIGITS dataset. Accuracies are shown both for 2D and 50D representations. Multimodal embeddings from the proposed model outperform unimodal and fusion approaches without importance-based weighting

| Model | 2D | | | 50D | | |
|---|---|---|---|---|---|---|
| | overall | image noise | speech noise | overall | image noise | speech noise |
| IMA (importance weights) | **83.88** | **69.29** | 82.20 | 96.94 | 86.99 | 94.46 |
| IMA (no importances) | 79.88 | 59.86 | 63.13 | 96.84 | 86.54 | 93.61 |
| IMA (unimodal image) | 76.90 | 12.06 | **83.05** | 86.92 | 9.41 | **95.31** |
| IMA (unimodal speech) | 62.26 | 66.00 | 10.80 | 82.88 | 90.80 | 11.70 |
| JMVAE-KL Suzuki et al. (2016) | 53.06 | 23.24 | 14.40 | 97.9 | 93.85 | 93.22 |
| MVAE Wu & Goodman (2018) | 41.32 | 39.69 | 10.38 | **98.18** | **94.95** | 92.37 |

Table 6: Emotion recognition accuracies for the best model on the validation set achieved by IMA model on IEMOCAP dataset, along with JMVAE-KL and MVAE baseline models.

| Model | Verbal | | Acoustic | | Multimodal | |
|---|---|---|---|---|---|---|
| | Unwt | Wt | Unwt | Wt | Unwt | Wt |
| IMA | 49.47 | 53.33 | 58.80 | 58.32 | 59.37 | 60.57 |
| JMVAE-KL | 53.43 | | 60.37 | | 61.17 | |
| MVAE | 54.96 | | 58.03 | | 62.63 | |

## A.5 DOWNSTREAM CLASSIFICATION EXPERIMENTS

The quality of the joint multimodal representation **z** learnt by the model is quantitatively evaluated by downstream digit and emotion classification tasks respectively from the MNIST-TIDIGITS and IEMOCAP datasets. We use a two-layer MLP (Multi-layer Perceptron) classifier with ReLU (Rectified Linear Unit) activation functions for the supervised tasks of emotion and digit classifications, given the representations learnt by the importance-based autoencoder and other baseline models. Table 3 contains details of classifier settings for each dataset. For MNIST-TIDIGITS, the classification is at digit level (each sample corresponds to a multimodal spoken and image digit pair), and for IEMOCAP it is at utterance level (where each sample corresponds to a spoken emotional utterance).

**MNIST-TIDIGITS :** Table 5 reports the classification accuracies on the digit classification task for both 2-D and 50-D multimodal representations for the importance based autoencoder and other baselines reported in Section 5.4. The accuracy on the overall evaluation set is reported along with that computed on subset of samples with uncorrelated noise (samples with synthesized image noise; and with synthesized speech noise - refer Section 4). In all cases the importance-weighted multimodal representations from IMA outperform those without importances.

In 2D, the importance-weighted representations from the IMA model outperforms the JMVAE-KL and MVAE baselines, as well the unweighted IMA representations. For higher dimensionality (50D), overall classification accuracy for the weighted IMA model is slightly less than other baseline approaches (including MVAE and JMVAE-KL). The unimodal encoders perform well as they have access to the other modality even in presence of noise. For example, the image encoder achieves an accuracy of 83.05% by utilizing the image digit even if the corresponding speech consists of noise, and vice-versa. Note that all classification accuracies are not comparable to state-of-the-art scores (¿99.5%) obtained on MNIST in contemporary literature Wan et al. (2013) due to no specialized convolutional/pooling layers and training on samples which include uncorrelated noise.

**IEMOCAP:** In Table 6, emotion recognition results on IEMOCAP are similar to the precision results reported in Section 6.2. After incorporating importances, verbal accuracy improves (absolute) by 3.86%, and multimodal accuracy improves by 1.2%. The IMA model outperforms achieves competitive performance to the baselines JMVAE-KL and MVAE for this task of emotion recognition.

