# OpenReview forum: "Importance-based Multimodal Autoencoder"
_ICLR.cc/2021/Conference — Reject_

### Official Review · AnonReviewer4 · 2020-10-26
**A good paper but some details are missing**

**Rating:** 7
**Confidence:** 4

**Review:**

Summary:
This paper tackles the problem of multimodal representation learning by proposing an importance based multimodal autoencoder (IMA). The advantages of their proposed IMA are two folds: it is flexible to remove some modalities, as the loss function is easy to include any combination of observations; it allows to learn modality specific features along with the shared latent representations.

Pros:
1. This paper is well-written and easy to understand.
2. The idea is novel. They addressed the important problems of multimodal representation learning, which could have wide applications in multimedia, computer vision and speech/emotion recognition context.
3. The experiments are extensive, and can support their conclusions.

Cons:
Some details are missing:
1) In Equation (2), what are lamda(align), lamda(rec), lamda(glob)? How do you choose them in your implementations? Are they hyperparameters?
2) What parameters you choose to visual the t-SNE of representations in Figure 1? As I know, the parameters inside t-SNE determine the entire visualization process.
3) In Section 5, why do you use two stages process to train your model? in the first stage, when the importance network is kept constant, does the network output a constant number (zero or one or other number)?
4) I would suggest to have a algorithm figure to demonstrate the relations between different network modules and loss functions.

---

> ### Author Response · Authors · 2020-11-16
> **We will be updating the rebuttal revision with additional details as suggested**
>
> We thank Reviewer 4 for the constructive comments. Regarding the details which are missing, we will be updating them in the rebuttal revision as described below:
> 1. $\lambda_{align}, \lambda_{rec}$ and $\lambda_{glob}$ are the associated hyper-parameter weights for the alignment, reconstruction and global multimodal regularization terms which appear in Equation (2). Since it is not straightforward to tune such parameters for an unsupervised model, we start with equal weights for all losses and observe if that is sufficient to minimize all terms simultaneously during training. For the MNIST-TIDIGITS and IEMOCAP experiments, we have found that equally weighing them ie. $\lambda_{align} = \lambda_{rec} = \lambda_{glob} = 1.0$ works.
> 2. For t-SNE we use the BHSNE implementation with parameter values: perplexity = 10.0 and theta = 0.5. The t-SNE parameters (such as perplexity) do determine the visualization process as the reviewer has mentioned. However, t-SNE is also known for its ability to effectively represent the proximity between samples in a lower dimensional space. Thus we believe the high-importance regions (for example in Fig 1, 3, and 4) would still be dimensionally reduced to a coherent cluster even for different t-SNE settings as the underlying IMA representations have this property.
> 3. The multimodal autoencoder and importance networks do not have compatible objectives. Minimizing the cross-correlation loss would identify the important regions of each modality but using them as weights in the multimodal autoencoder would not necessarily minimize reconstruction loss. Learning the importance weights directly from the multimodal autoencoder alone by minimizing reconstruction loss would still weigh the low-importance regions for the sake of modality reconstruction, but not minimize the cross-correlation loss. We are effectively searching for the equilibrium of these objectives, which is more suitable with an alternating minimization stage. Experimentally as well, we have found it practically inconvenient to optimize all the autoencoder and importance losses together and ensure they all minimize and converge. Doing a two-stage alternating minimization process mitigates this issue.
> In the first stage, when the importance network is frozen, the importance outputs are held constant for each sample during autoencoder training, however they vary across multimodal samples depending on the importance network output.
> 4. We thank Reviewer 4 for the suggestion to have an algorithm/overview figure explaining the different network components and losses, and will be inserting it into the uploaded rebuttal revision. This has also been suggested by the other reviewers.

---

### Official Review · AnonReviewer1 · 2020-10-27
**Official Blind Review #1**

**Rating:** 5
**Confidence:** 5

**Review:**

The paper proposes the IMA model, a scalable model that learns modality importances and robust multimodal representations through a novel cross-covariance based loss function. The proposed model performs unimodal inference in absence of modalities and also addresses the problem of detecting important subspaces in each modality through weighted cross-covariance loss terms, which are minimized by unimodal importance networks. Results are shown that the IMA model is able to distinguish digits from uncorrelated noise, and word-level importances are learned that correspond to the separation between function and emotional words. The multimodal representations learned by IMA are also competitive with state-of-the-art baseline approaches on downstream tasks.


Pros:

+ Importance-based Multimodal Autoencoder is a very interesting topic, Importance Network Training also would benefit to other fresh ideas and new approaches.
+ I really like the idea of learning modality importances and robust multimodal representations through a novel cross-covariance based loss function. I agree with the authors that this should help a lot for unimodal inference in absence of modalities and also addresses the problem of detecting important subspaces in each modality.


Cons:

- While I like the premise of the paper, I feel that it needs more work. My main concern is that seeking to learn weights $y_{ij}$ (the importance of each sample $x_{ij}$) should not be equivalent to the degree to which $x_{ij}$ does not belong to $R_j$. In other words, if the goal of *importance network* is only filtering uncorrelated noise, how is the importance reflected? Perhaps this should be called correlation. My understanding is that the importance network should be quantified to a weight range, similar to the attention mechanism. If not, the author should make a more clear explanation of the *importance network*.

- The text is quite hard to read, there are many places in the paper that are not explained clearly, especially the related work. This section more like a pile of other related works but lacks coherence and puts the proposed method into context.

- The model description section would be easier to understand if the author can add an overall framework figure, especially *Importance Network Training* need a more detailed description.

- At the *Importance Priors* section, what is the $L_{local}$? At *Dataset IEMOCAP* section, the author stated "We also remove the MFCC features, and thus most of the acoustic variability within the utterance is removed resulting in factors of variation (such as emotion) which are more global in nature." What is *global in nature*? What is the difference between defining local and global?

- The experimental part is a bit confusing and needs more analysis, why is the result of Table 1 worse than those two models: JMVAE-KL and MVAE?

- This paper hardly cited references from 2019 and 2020, and it is recommended that the author pay attention to some updated work.

-  The labels of the figures and tables in the paper should preferably appear and quote in order.

---

> ### Author Response · Authors · 2020-11-16
> **Author Replies to Reviewer 1**
>
> We thank Reviewer 1 for the constructive comments and would like to present the following points as replies.
> 1. To better explain the main premise of the paper, we refer to the shared-private assumption frequently employed in multimodal modeling, and also assume that the multimodal representation $z$ is a factor underlying all modalities. Thus, within each modality, we assume there is a (shared) factor contributing to and correlated with the multimodal latent factor and a (private) uncorrelated factor. Learnable Importance $y_{ij}$ is defined as the degree to which the shared factor dominates, i.e., how much j-th sample $x_{ij}$ in each i-th modality correlates with $z_j$. The importance network learns the decision boundary in each modality to partition the modality space into regions which are (and are not) correlated with $z$. In the emotion recognition example, a neutral function word (“I”) spoken in a sad tone would still map to a multimodal representation of sadness. How do we know the word “I” is not important here for inference? This can be learnt from a multimodal corpus by observing that the occurrence of the word  “I” is not correlated with the emotion of the utterance (i.e., occurence of “I” is throughout the acoustic space instead of clustered with only the sad utterances).
> 2. Regarding the readability of the paper, we can certainly improve the overall layout of the paper and will be uploading it in the revised submission. The figures and tables will be referenced in the order of their appearance. In the Related Works section, we will be organizing the references by research domain and also cite relevant papers from recent years (2019, 2020). We will be adding an overall framework figure to the paper.
> 3.  For IEMOCAP, we utilize acoustic features which incorporate multiple factors of variation such as phonetics (acoustic characteristics pertaining to what is said) as well as extra-linguistic factors (acoustic characteristics pertaining to how or by whom it is said) such as emotions (happy vs. angry voice, etc.), loudness, tone of voice, and gender. While all of these latent factors influence the observed acoustic features, they are both local as well as global in nature. Local factors of variation are transient, for example, phonetics (fricative, vowel sounds, etc.) and pause fillers. Global factors of variation influence the whole utterance, for example higher or lower average pitch (which can vary with gender, i.e., a global attribute). The MFCCs mostly capture phonetics, and we have removed them as we do not wish to learn how words correlate with their phonetics, but rather to their emotional/extra-linguistic attributes.  We have found that IMA can learn the correlations between words and their
> 4. In the Importance Priors section, $L_{local}$ is the loss-term which represents the KL- divergence between (a) a distribution representing prior knowledge of where the importance prior $\rho$ is expected to lie in range [0, 1] and (b) a distribution over space {important, non-important} of the fraction of samples in each modality predicted to be important by IMA. Overall, this is a regularizing prior over the importance network predictions and penalizes the importance training if the predictions diverge from the prior knowledge of how important samples in each modality are.

---

### Official Review · AnonReviewer3 · 2020-10-27

**Rating:** 6
**Confidence:** 4

**Review:**

In this paper, the authors are motivated by two properties that are often missing from multi-view or multimodal algorithms, namely robustness to missing modalities at test time, as well as learning the relevance of modality/samples to the so-called shared latent space in an unsupervised manner.

To do so, the authors first consider that the shared latent space z results from a weighted combination of the outputs of M encoders corresponding to M different modalities.  This appears to be a convex combination (Eq. 1), although it is not mentioned that each weight should be positive explicitly (authors can clarify this in an updated version).  The shared space is also regularized with a squared l2 norm.

The authors subsequently force z to be close to the encodings of each modality (u(x)_j) while also reconstructing each observation from the same modality.  The authors introduce two more loss functions as a way to learn importance weights for samples/modalities (_{ij}).  This is done by a cross-correlation loss that learns high importance when there is high correlation between modality/sample encodings and the shared latent space.    This is also influenced by a hyperparameter indicating prior knowledge on data quality per modality.  The experimental results demonstrate that the importance mechanism improves results, at least for the specific datasets and experimental setting.

Some comments follow below:

-> generalization and robustness:  As an importance is assigned to each sample per modality, what happens when outliers are present in the data?  Could this mechanism essentially overfit such noise (perhaps common in all modalities) due to the individual sample weighting, and lead to a lack of generalization on an unseen dataset?  Robust methods (e.g., depending on l1 regularization) might be able to cope with such scenarios better


-> missing modalities: the authors highlight that the method can handle missing modalities - by setting the importance weight for the corresponding modality to zero.  However there is not enough experimental evidance (unless I have missed something) in the main paper to demonstrate the performance of the proposed method in this respect. Also, models that impose a shared-space assumption (in the most simple way, shared-siamese style encoders leading to common layers) can also handle missing modalities in the same manner - so I wouldn't say that this is the strongest contribution of this paper.


->  The cross-covariance loss function (and in general, soft orth constraints) have been used in the context of private-shared space models (e.g., stemming from the classical Inter-Battery Factor Analysis).  In many ways, it would be interesting to review and link this literature with work dealing with discovering shared-representations (e.g., Deep CCA, Andrew ICML 2013), as this work could potentially be considered an extension of these methods.

-> attention mechanisms are not mentioned in the paper, however Eq. 1 is reminiscent of an attention mechanism.  Could we consider this approach as a form of attention-autoencoder, where the attention weight is used to indicate importance?

-> was the prior hyperparameter that controls the number of corrupted samples tuned in some automatic way? (e.g., grid-search). Of course the features are extracted in an unsupervised manner, however (i) the samples are corrupted by artificial rather than real noise in the experiments and this is known during the experiment, and (ii) in unknown test-sets, the noise distribution might vary and could be unknown.  For a particular choice of hyper-parameter values, the authors state that  this "indicates that these would be the optimal values for best performance in image and speech modalities respectively" - but can we really be confident that these values will be consistently better?  Could this just be fitting the particular dataset properties and harming generalization?

-> optimization: why is the particular approach employed with alternating optimization?   It is difficult to claim convergence for the given setting, but have the authors observed good convergence properties in the model?  What values were given to hyperparameters \lambda for each experiment?  Also, it appears that there are several constraints on z - some countering each other in some way.

-> continuing from the previous point, for example z has to be equal to each u(x)_j, but also equal to a convex combination of all u(x)_js, while an additional cross-covariance loss is imposed so that z correlates with u(x)_j.  At the same time, u(x)_j must reconstruct each x_j - which essentially means to my understanding that in an optimal setting, the decoder should contract or absorb all information that is not 'shared' amongst the modalities (what the authors call uncorrelated noise).  Does this cause additional instability during training?  Could it be that information inherent only to one modality is still useful, and not uncorrelated noise - which however would be disregarded within this approach?

---

> ### Author Response · Authors · 2020-11-17
> **We have clarified and explained major points of the paper for Reviewer 3**
>
> We thank Reviewer 3 for the constructive comments and would like to present the following points as replies. One point we wish to make here is the definition of noise in the context of our paper. For our experiments, noise is uncorrelated, i.e., the corrupted region in each modality that has the property of not being well-correlated with the other modalities. The detection of noise here is not through outlier detection (as in anomaly detection), but is learnt from the pattern of correlation with other modalities by the importance network.
> 1. We have not experimented with robustness to outliers, however if the outlier noise in a modality is not correlated with the other modalities (and thus would have low correlation with multimodal z), then it would be expected to be uncorrelated noise, and detected and filtered out by the importance network. Regarding each sample being provided an importance weight, this is constrained by a decision boundary learnt by the importance network. As long as the network is not over-parameterized and is tuned to learn well-formed boundaries, it should be able to learn proper importances without overfitting.
> 2. We agree with the reviewer that the shared-space assumption is also being utilized in this paper to enable robust inference in absence of modalities (similar to the example provided by the author). We will be augmenting Table 2 to report additional performance metrics when only one modality is provided (either image/speech digits). In the emotion retrieval experiments in Table 1, the “Verbal” and “Acoustic” columns already report performances when only one modality is provided. We are also not claiming missing modality robustness as the main contribution here but rather: (1) scalability through fewer loss terms to optimize during training compared to JMVAE-KL and MVAE and (2) the detection of important regions in each modality.
> 3. In our Related Works section, we shall certainly link our proposed IMA model as extensions of the previous work the reviewer has mentioned (IBFA as well as Deep CCA).
> 4. We are drawing a distinction between attention mechanisms and the concept of importance here in this paper. This is mainly due to two reasons. Firstly, the classical attention mechanisms in seq2seq or image-text applications focus on learnable attention weights within the input sequence during decoding. This is intra-modality, in contrast to importance weights with which we weigh different modalities in IMA. Secondly, importance is mostly based on learning which regions in each modality are correlated with other modalities, in contrast to standard attention mechanisms which are based on learning regions of an input sequence to which a decoder can focus on when considering a specific token within the  output sequence.
> 5. There was no specific automatic method to tune the importance prior, considering that the ground truth digit/emotion labels are not available during training of the autoencoder or importance networks. However, during evaluation we do compare the performance against these labels to obtain an insight into the performance at detecting uncorrelated noise. In Fig A3 (appendix), the high F1 score obtained during these comparison experiments do indicate our confidence that at a certain value of importance prior, we do successfully identify the important regions in each modality. We will be revising our claim in the paper that “..indicates this would be the optimal values...”  In the experiments on the MNIST-TIDIGITS dataset, the uncorrelated noise was artificially synthesized, however for the emotion understanding experiments on IEMOCAP, we have not synthesized any corruption. The dataset naturally has uncorrelated noise, for example function words in the verbal modality are not correlated with the utterance emotion and can be detected by our proposed model.
> 6.The multimodal autoencoder and importance networks do not have compatible objectives. Minimizing the cross-correlation loss would identify the important regions of each modality but using them as weights in the multimodal autoencoder would not necessarily minimize reconstruction loss. Learning the importance weights directly from the multimodal autoencoder alone by minimizing reconstruction loss would still weigh the low-importance regions for the sake of modality reconstruction, but not minimize the cross-correlation loss. We are effectively searching for the equilibrium of these objectives, which is more suitable with an alternating minimization stage. Experimentally as well, we have found it practically inconvenient to optimize all the autoencoder and importance losses together and ensure they all minimize and converge. Doing a two-stage alternating minimization process mitigates this issue.
>
> The hyperparameters $\lambda$ have been set to 1.0 for the multimodal autoencoder in all experiments, and we have found this to be satisfactory in reducing the loss terms controlling z.

---

> > ### Author Response · Authors · 2020-11-25
> > **We have clarified and explained major points of the paper for Reviewer 3 (Part 2)**
> >
> > To continue addressing the reviewer concerns, we would like to also note the following :
> > - The competing nature of the multimodal autoencoder and the importance networks as mentioned above has a parallel with GANs (Generative Adversarial Networks) where the generator and discriminator networks optimize conflicting objectives through an alternating training. The objective here is to focus on finding an equilibrium point rather than a global optimal point.
> > - In an optimal setting if the j-th modality sample $x_j$ belongs to a region of uncorrelated noise, $z$ would not incorporate $u(x_j)$ in the convex combination, and the reconstruction loss would be high, and the alignment loss would be high as well. The decoder does not reconstruct $x_j$ from $u(x_j)$ but rather from $z$ and thus would not absorb the private information from $x_j$. To achieve that, we would need $x_j$ to also have its own autoencoder to absorb the private information, which would not be scalable for multiple modalities. We do not expect this to affect convergence within a single round of autoencoder training, as the importance network is not being optimized simultaneously; even though it would converge to a higher loss value.
> > - It is possible that there could be private information in each modality which would help the multimodal inference. However we would consider that outside the scope of our paper, and consider only a notion of importance in each modality which does not take into account this private factor. Emotion understanding from multimodal data is an example where this definition would prove useful, which provides the setting for our experiments.
> > - There are also parallels between the IMA model losses and a multimodal VAE with shared and private latent factors. Consider $z$ as the multimodal latent factor, $u_j$ to be the private factor for modality $j$ and $y_j$ to be the binary indicator of importance. In that case, $log P(z, u, y, x) = \log P(x|z,u,y) + \log P(z,u) + \log P(y)$ when the  indicator and latent factors are all mutually independent. The ELBO component on the first term would be optimized by a network similar to the multimodal autoencoder. The ELBO for the second would match $P(z,u)$ with the posterior $q(z, u|x)$, which would enforce zero-mean on $u$ and $z$, along with enforcing zero correlation between them. The ELBO component for $P(y)$ is functionally similar to the $L_{local}$ where the importance network output is matched to a prior $P(y)$ through a KL-divergence.

---

### Official Review · AnonReviewer2 · 2020-10-29
**Elegant unsupervised importance based auto-encoding model - good ideas and some early results - needs more experiments and analysis**

**Rating:** 6
**Confidence:** 3

**Review:**

This paper presents a multimodal Autoencoder framework that learns the multimodal latent representations alongwith the importance of regions in each modality’s representation space in an unsupervised fashion. Multimodal fusion algorithms either use complex architecture representations or use disentangling joint representations for improving generative auto-encoding architectures using VAEs, GANs, WAE and some variants of these. This paper presents an elegant importance based model and architecture that takes into account various local and joint loss functions along with alignment factors to represent the Autoencoder model.
Some questions/comments that would make the paper more readable:
- Architecture diagram: Please provide an architecture diagram for the model to help the readers
- How good are the embeddings for downstream applications: Can we use these representations and compare performance with other learnt embedding models? How would the performance be with missing modalities?
- Table 1 - It is surprising to see that weighted Precision is less than unweighted Precision - any thoughts?
- Comparison to other Autoencoders: How does this compare to denoising Autoencoder? How does this compare to Wasserstein autoencoders and the variants such as the multimodal factorization model neural architecture?
Figure 2: Any relationship or companion to self-attention weights and word level importances
- Number of network params compared to MVAE model that the authors have compared to? How fast is the model training compared to other similar models?

---

> ### Author Response · Authors · 2020-11-16
> **We will be updating the results for downstream tasks/baselines, as suggested by the reviewer**
>
> We thank Reviewer 2 for his/her constructive comments. Our replies to the reviewer comments are provided below:
> 1. As also requested by the other reviewers we will certainly update the revised submission with an overall architecture diagram.
> 2. Due to page limit we did not mention expansive experiments on evaluating quality of these embeddings for downstream tasks in the main paper, however we have presented results utilizing the learnt embedding for multimodal digit and emotion recognition experiments in Appendix A4. The IMA model is compared to the MVAE and JMVAE-KL models and the performance of IMA is competitive with these baselines. We also include performances in the case of missing modalities.
> 3. The multimodal embeddings/importances are learnt by IMA without access to the emotion labels during training, and thus it is possible that in Table 1, the importance network for the speech modality has inferred certain frames in the acoustic modality as uncorrelated noise and thus not important, whereas given the emotion labels they might be relevant. For those samples, it is possible that an unweighted representation (where all frames are considered - simple average) might be providing better precision performance than weighted representations (where the frame representations are weighted by importance) when the emotion label is considered as a ground truth during evaluation. We feel this is more due to mismatch between training and evaluation objectives, and that the lower precision for the acoustic modality is also influencing the multimodal performance.
> 4. Time permitting, we will be comparing IMA and mentioning the results to some of the models suggested by the reviewer (ie. denoising autoencoder / Wasserstein autoencoder). Could the reviewer provide a link to the citation for the multimodal factorization model? Regarding Fig 2, the word-level importances are based on correlations between words and the acoustic frames they co-occur with, which is different from self-attention weights which are based only within the words themselves. Further self-attention carries a semantic meaning, compared to IMA importance-weights at word level, which we have found relates to emotion.
> 5. In comparison to the MVAE model, we have ensured that the encoder and decoder sizes are the same as those for IMA to obtain a comparable experimental setting for benchmarking. We will be inserting a section in the Appendix on model complexity comparison between IMA and baseline models. For the MNIST-TIDIGITS experiments (50D), MVAE has a total of 783000 parameters for the encoder and 1217360 parameters for the decoder, amounting to ~2 million parameters. In comparison, IMA has 753000 parameters for the encoder and 754280 parameters for the decoder, adding to ~1.5 million parameters. The importance network has 723402 additional parameters. We have found model training time for IMA to be faster than both JMVAE-KL and MVAE due to it having a simpler architecture than the baselines (fewer loss terms to optimize and there is no probabilistic assumption, and thus no covariance calculation unlike VAE). The importance networks do incur overhead, however they can be parallelized across modalities. We have also experimented with weight-sharing between encoders and the importance network, however do not obtain good convergence.

---

### Author Response · Authors · 2020-11-22
**Update about Rebuttal Revision**

To address the reviewer comments, we have uploaded a new revision to our paper. The following are changes we have made to the paper:
- We have added an overview figure (an overview architectural diagram) of the IMA model, including the loss terms. This is in Figure 1
- We have rewritten the Prior Works section to include some additional citations to foundational (ie. IBFA, DeepCCA) and recent work from 2019 and 2020. We have also drawn connections between our proposed model and these prior approaches.
- We have expanded out Table 1 to also include precision results when one modality (images/speech) is missing. Under those settings as well, the performance of IMA model is competitive with the competing baselines.
- In the appendix section, we have clarified acoustic feature extraction (ie. MFCC filtering), and also introduced Section A.4 to list the number of parameters in IMA and the competing baseline models.

---

### Author Response · Authors · 2020-11-25
**Rebuttal Revision v2**

We have uploaded a new rebuttal version which includes denoising autoencoder performances on the MNIST-TIDIGITS dataset and also rephrases our claim in Section 6.1 / MNIST-TIDIGITS ("... which are empirically the suitable
priors for the importance network predictions to agree with external labels in the image and speech
modalities respectively...")

---

### Decision · Program_Chairs · 2021-01-07
**Final Decision**

**Decision:**

Reject

**Comment:**

The paper proposes an auto-encoder framework IMA, a scalable model that learns the importance of modalities along with robust multimodal representations through a novel cross-covariance based loss function, in an unsupervised manner. They have compared their approach to SOTA methods via multiple experiments and shown how IMA gives better performance.

The authors have addressed some of the reviewers' feedback. However, as pointed out by the reviewers, the experimental section needs better analysis of results and comparison to other methods, and the modeling section needs to be better explained and motivated. The authors have made changes in their revision, however the ICLR review process does not allow for checking the camera-ready. Since we cannot accept the paper in its current form (or with small variations) and there have been many competitive submissions, we would encourage the authors to make their revisions and resubmit to other venues.